# Effect of Prenatal Yoga on Heart Rate Variability and Cardio-Respiratory Synchronization: A Prospective Cohort Study

**DOI:** 10.3390/jcm11195777

**Published:** 2022-09-29

**Authors:** Ivan Žebeljan, Miha Lučovnik, Dejan Dinevski, Helmut K. Lackner, Manfred G. Moertl, Izidora Vesenjak Dinevski, Faris Mujezinović

**Affiliations:** 1Department for Women’s Health, Health Center Lenart, 2230 Lenart v Slovenskih Goricah, Slovenia; 2Department of Perinatology, Division of Obstetrics and Gynaecology, University Medical Centre Ljubljana, 1000 Ljubljana, Slovenia; 3Medical Faculty, University of Ljubljana, 1000 Ljubljana, Slovenia; 4Medical Faculty, University of Maribor, 2000 Maribor, Slovenia; 5Otto Loewi Research Center, Section of Physiology, Medical University of Graz, 8036 Graz, Austria; 6Alpen-Adria University Klagenfurt, 9020 Klagenfurt, Austria; 7Sončna Vila Yoga Studio, 2000 Maribor, Slovenia; 8Department of Perinatology, University Medical Centre Maribor, 2000 Maribor, Slovenia

**Keywords:** pregnancy, yoga, heart rate variability, cardio-respiratory synchronization, autonomic nervous system

## Abstract

The objective was to assess the effects of prenatal yoga on heart rate variability (HRV) and cardio-respiratory synchronization, used as proxies of autonomic nervous system activity. Sixty-nine healthy pregnant women were included; 33 in a yoga group attending at least one 90-min yoga class weekly throughout pregnancy, and 36 controls not involved in formal pregnancy exercise programs. Measurements of the time domain (SDNN, standard deviation of regular R-R intervals, and RMSSD, square root of mean squared differences of successive R-R intervals) and frequency domain (ln(LF/HF), natural logarithm of low-frequency to high-frequency power) HRV indices, as well as cardio-respiratory synchronization indexes were performed once per trimester before and after yoga or 30-min moderate-intensity walk. A statistical comparison was performed using a three-way analysis of the variance (*p* < 0.05 significant). Both the time domain and frequency domain HRV indices showed significant shifts towards parasympathetic dominance following yoga when compared to the controls throughout pregnancy (*p* = 0.002 for SDNN, *p* < 0.001 for RMSSD, and *p* = 0.006 for ln(LF/HF), respectively). There was a statistically non-significant trend towards higher synchronization between respiratory frequency and heart rate following yoga vs. controls (*p* = 0.057). Regular prenatal yoga was associated with enhanced parasympathetic activation persisting throughout pregnancy.

## 1. Introduction

Regular and adequate physical activity during pregnancy has positive health effects for pregnant women and their foetuses [1,2]. It has been proven to reduce the incidence of gestational diabetes, hypertensive disorders of pregnancy, urinary incontinence, and pregnancy-related pelvic girdle pain [3,4,5,6]. Physical activity during pregnancy also has beneficial impacts on labour and it contributes to a faster postpartum recovery [1,2,7].

Yoga is a psycho-physical exercise that includes physical postures (asanas) and breathing (pranayama), concentration (dharana), and meditation (dhyana) techniques. It is based on the ancient Indian heritage and philosophy [8]. Since the early 20th century, it has also become increasingly popular in the West. Around 6.6% of adults in the US and 20.5% of adults in the UK have practiced or are practicing yoga [9,10]. Approximately 70% of yoga practitioners are women. The majority of them are of reproductive age [11]. Consequently, a rising number of pregnant women are also practicing yoga [11,12].

A growing body of evidence suggests potential beneficial effects of practicing yoga during pregnancy. Studies showed a positive impact of prenatal yoga on maternal psychological health, pregnancy-related pain, as well as several perinatal outcomes, such as a reduction in preterm birth, hypertensive disorders of pregnancy, and gestational diabetes [11,13,14,15,16]. It is, however, still not known whether beneficial effects of prenatal yoga are superior to those of other forms of aerobic physical exercise, such as walking. Moreover, the exact mechanisms by which yoga could improve perinatal outcomes have not yet been elucidated [17,18].

As many of yoga’s positive effects on health are thought to be mediated through its effects on the autonomic nervous system, we designed a study to assess the effects of yoga practice throughout pregnancy on heart rate variability (HRV) and cardio-respiratory synchronization, used as proxy measures for autonomic nervous system activity. We also compared effects of prenatal yoga on the autonomic nervous system to those of moderate walking.

## 2. Materials and Methods

### 2.1. Participants

Sixty-nine healthy pregnant women were included in the study. The exclusion criteria were as follows: multiple pregnancy, cardiovascular disease (including hypertension and arrhythmias), taking medications that would affect heart rate or blood pressure, psychiatric disorders, epilepsy, kidney disease, liver disease, known fetal anomaly, rheumatoid autoimmune disorders, diabetes mellitus, and alcohol and/or illicit drug abuse. All women were of Caucasian ethnicity. Study participants were divided into two groups: yoga group and controls. 

Participants in the yoga group were recruited during the first trimester of pregnancy at Soncna vila yoga studio in Maribor, Slovenia and during regular prenatal visits at the Department of Perinatology of the University Medical Center Maribor and other health centers in the area. Women were instructed to attend one prenatal yoga class per week. We initially recruited 44 pregnant women in the yoga group. Eleven were excluded as three stopped practicing yoga, seven did not complete all measurements, and one developed arrhythmia during pregnancy, leaving 33 participants in the yoga group for analysis. 

The control group consisted of healthy pregnant women attending regular prenatal visits at the Department of Perinatology of the University Medical Center Maribor and other health centers in the area. They were also recruited during the first trimester of pregnancy. Only women not attending any formal prenatal exercise program were offered entrance in the study as controls. Forty women were initially recruited in the control group. Four were excluded due to not completing all measurements, leaving 36 participants in the control group for analysis.

The baseline characteristics of the participants are shown in Table 1. Maternal height and body mass index (BMI) were similar in the two study groups. Maternal age was higher in the control group, while the proportion of nulliparous women was higher in the yoga group. 

Informed consent was obtained from the participants, after they received a detailed explanation of the study. The study was performed in accordance with the 1964 Declaration of Helsinki in its current form and was approved by the authorized ethics committee (Slovenian National Medical Ethics Committee; Project number 0120-575/2018/5, approved on 22 February 2019).

### 2.2. Study Design and Procedure

This prospective cohort study was performed at a single yoga studio in Maribor, Slovenia, from August 2020 to April 2022, in collaboration with the Department of Perinatology, Division of Gynecology and Perinatology, University Medical Center Maribor, Slovenia, the Department of Perinatology, Division of Obstetrics and Gynecology, University Medical Center Ljubljana, Slovenia, and the Otto Loewi Research Center, Section of Physiology, Medical University of Graz, Austria. The study is registered at clinicaltrials.gov with the identifier NCT04476368.

Participants in the yoga group attended one guided yoga class per week. Classes consisted of pregnancy-adapted yoga practices according to the system Yoga in Daily Life [19]. They lasted for 90 min and consisted of an initial relaxation (10 to 15 min), followed by yoga postures (asanas) and stretching exercises (45 to 60 min), and concluding breathing (pranayama), concentration (dharana), and meditation (dhyana) techniques (20 to 30 min). The adaptation of yoga practices for specific gestational age was based on in-depth consultations with gynecologists and experienced physical therapists. A single certified yoga instructor with 25 years of experience led all yoga classes. Measurements were performed before and after yoga practice once per trimester of pregnancy. 

Measurements in the control group were performed before and after a 30 min moderate intensity walk once per trimester of pregnancy.

### 2.3. Data Acquisition and Preprocessing

#### 2.3.1. Data Acquisition

All measurements were performed in the afternoon (approximately the same time of day). Women were not eating and consuming caffeine one hour prior to measurements. The measurement protocol consisted of an adaptation period and 10 min recording at rest. Continuous monitoring of blood pressure (sampling rate (sr) = 100Hz, blood pressure range = 50–250 mmHg, ±5 mmHg), and heart rate (R-R intervals derived from 3-lead electrocardiography (ECG), sr = 1 kHz, f_cut-off_ = 0.08–150 Hz) and thoracic impedance (sr = 50 Hz, Z_0,range_ = 10–75 Ω) were carried out with the Task Force^®^ Monitor (CNSystems, Medizintechnik AG, Graz, Austria) [20]. Continuous blood pressure was measured on the proximal limb of the index or middle finger by a refined version of the vascular unloading technique and corrected to absolute values with oscillometric blood pressure measurements on the contralateral upper arm by the Task Force^®^ Monitor. The method used to derive continuous non-invasive blood pressure measurements has been previously described and validated [21]. Raw data were export to MATLAB^®^ (The Math Works, Inc., Natick, MA, USA) data format for further analysis.

#### 2.3.2. Artifact Handling

To check the quality of the physiological data and calculate the interbeat interval time series, we have been consistently using for several years a semi-automatic artifact-handling software developed by our research group [22,23]. In brief, its main criteria for the ECG artifact handling were: 

the pattern of the QRS complex and the time of occurrence within the ECG to identify “ectopic beats”

physiological limits on an individual, age-depending basis, andthe maximal percentage of change in relation to the standard deviation of the signal.

Our algorithm used the R-R intervals and had the additional advantage of incorporating the whole information of the ECG. In the next step, the continuous blood pressure values were checked based on the corrected ECG and R-R interval, respectively, as well as physiological and technical limits.

Physiological limits and the percentage of change were calculated on an individual basis based on the envelope of the R-R intervals, systolic, and diastolic blood pressure intervals, respectively, using the Hilbert transformation. To obtain the interval time series with equidistant time steps for the calculation of frequency-domain and synchronization variables, the beat-to-beat values were resampled at 4 Hz, using piecewise cubic spline interpolation after the described semiautomatic artifact correction.

### 2.4. Analysis Procedure

For the calculation of the variables, a 120 s interval at rest following the adaptation phase was chosen in order to ensure the stability and stationarity of the signal, considering that the segment should be no less than approximately ten times the wavelength of the lower frequency bound of the investigated component.

#### 2.4.1. HRV Variables 

All HRV variables were calculated and defined following the recommended guidelines [24].

The time domain variables, i.e., SDNN, the standard deviation of the regular R-R intervals, and RMSSD, the square root of the mean squared differences of successive R-R intervals, were calculated for the 120s segments.

For the frequency domain variables, Power Spectral Density (PSD) estimates were calculated from the R-R intervals via Burg’s method (model order 24) after removing the trend (2nd-order). Low frequency (LF) was defined as 0.04–0.15 Hz, high frequency (HF) was defined as 0.15 Hz or above. Due to the skewed distribution of frequency domain variables, a natural logarithmic transformation was applied, represented in ln(LF/HF).

#### 2.4.2. Blood Pressure Variables, Respiratory Frequency, and Phase Synchronization Variables

For the blood pressure variables, mean values were calculated for systolic, mean arterial, and diastolic blood pressure.

For the calculation of the respiratory frequency, the respiratory signal was derived from the thoracic impedance and the signal was band-pass filtered (butterworth filter of 4th order, range 0.05 to 0.8 Hz). The resulting signals were down sampled to 4 Hz to obtain corresponding times as R-R intervals and blood pressure time series. The respiration frequency (RF) was calculated as mean value using pattern recognition methods representing breathing cycles. 

For the calculation of the phase synchronization variables of R-R interval and blood pressure time series, as well as respiration, software developments of our research group were used [25,26]. In brief, the analysis of synchronization was based upon the weak coupling of two chaotic systems, whereas each oscillator could be described by its amplitude and phase as a function of time [25].

To admit a clear physical interpretation, which was given only for narrow band signals, we used band-pass filtered time series, representing high frequency components. We hypothesized that the phase synchronization of R-R interval, blood pressure, and respiration quantified by the synchronization index γ could be used for a more detailed analysis of the effect of yoga vs. controls, that is, e.g., the influence of respiration on the different branches of the autonomic nervous system.

### 2.5. Statistical Analysis

For a comparison of the baseline characteristics of participants in the two study groups (yoga vs. controls), Student’s t test was used for continuous variables and Chi-square test for categorical variables. For continuous variables, data were expressed as means with standard deviations. Categorical data were summarized as frequencies and percentages. Differences in all described outcome variables were statistically compared using 2 × 3 × 2 three-way analyses of variance. Time (before and after yoga or walk) and trimester of pregnancy (first, second, and third trimester) were the within-subject’s factors in these analyses, and intervention (yoga vs. control) the between-subject factor. These analyses allowed to determine whether the intervention (yoga vs. control) significantly influenced a given parameter measured when considering the physiological change in this parameter throughout pregnancy and following any physical activity.

For all tests, a two-tailed *p* value ≤ 0.05 was considered statistically significant. The software used for statistical analysis was IBM SPSS Statistics for Windows Version 25.0 (IBM Corp., Armonk, NY, USA).

## 3. Results

Table 2 presents HRV variables before vs. after yoga practice (yoga group) and before vs. after 30 min walk (control group) in all three trimesters of pregnancy. 

The between-subject analysis showed a significant decrease in heart rate following yoga practice compared to heart rate decrease following walking (controls) in all three trimesters (*F*(1,67) = 38.2, *p* < 0.001). We also observed a significant increase in time domain HRV parameters following yoga practice compared to controls regardless of pregnancy trimester (F(1,67) = 9.9, *p* = 0.002 for SDNN and F(1,67) = 17.1, *p* < 0.001 for RMSSD, respectively). In terms of frequency domain HRV variables, ln(LF/HF) was significantly decreased following yoga practice compared to controls throughout pregnancy (F(1,67) = 8.1, *p* = 0.006) (Figure 1). These results suggested that the higher “total” HRV (SDNN) following yoga compared to controls was most probably driven by a more pronounced parasympathetic activity represented by RMSSD and ln(LF/HF). 

Table 3 presents blood pressure variables, respiratory frequency, and phase synchronization variables before vs. after yoga practice (yoga group) and before vs. after 30 min walk (control group) in all three trimesters of pregnancy.

Compared to controls, SBP was significantly reduced following yoga practice in all trimesters of pregnancy (*F*(1,67) = 38.2, *p* < 0.001). We observed no significant differences in the changes in MAP (*F*(1,67) = 0.04, *p* = 0.835), DBP (*F*(1,67) = 0.58, *p* = 0.447), and RF (*F*(1,67) = 0.39, *p* = 0.534) after yoga vs. walking (control group). 

We found no statistically significant differences in synchronization variables between yoga and control groups (*F*(1,67) = 2.12, *p* = 0.150 for *γ_SBP x RR_*_,_
*F*(1,67) = 3.76, *p* = 0.057 for *γ_RF x RR_*_,_ and *F*(1,67) = 1.74, *p* = 0.191 for *γ_RF x SBP_*, respectively). However, there was a trend towards a higher *γ_RF x RR_* when comparing before vs. after yoga measurements with no such effects after moderate-intensity walking (*F*(1,67) = 3.76, *p* = 0.057).

## 4. Discussion

Both time domain and frequency domain HRV indices showed significant shifts towards parasympathetic dominance following yoga sessions compared to moderate intensity walking in healthy pregnant women throughout pregnancy. 

Several studies have examined the effects of yoga on autonomic nervous system outside of pregnancy [27,28,29]. HRV indices showed significant acute shifts in autonomic balance towards vagal dominance following a single yoga session in experienced yoga practitioners and non-experienced office workers [30,31,32]. However, chronic effects of yoga practice on HRV remain inconclusive due to substantial heterogeneity of studies [27]. Data on effects of yoga in pregnancy on autonomic nervous system are limited to one study from Bangalore, India [18]. Satyapriya et al. showed an acute decrease in sympathetic tone (decreased low frequency (LF) band power and LF/high frequency (HF) ratio) and an increase in parasympathetic tone (HF band power) in both yoga and standard prenatal exercise group [18]. HRV changes were, however, more profound following yoga vs. standard exercise. Our results were in line with these findings. We have demonstrated that short-term shifts in autonomic balance to the parasympathetic branch of the autonomous nervous system following yoga sessions in pregnancy can be replicated in a cohort of Western pregnant women. This is important, as a different perspective on practicing yoga in Indian society could have affected the comparison between yoga and other forms of physical activity in pregnancy. Satyapriya et al. showed no changes in HRV after 16 weeks of yoga training. This lack of long-term effect of yoga on HRV could have been due to several reasons. Participants in this study were randomly assigned to either yoga or standard prenatal exercise group. Therefore, their approach to yoga, motivation, and compliance with the exercise protocol could have been very different from women in our study, who have chosen to practice yoga by themselves. In addition, women in the Satyapriya study mostly practiced yoga alone at home, while in our study women attended weekly guided classes. This could have also impacted the motivation and attitude towards yoga practice throughout pregnancy. Moreover, prenatal exercises in the control group included both stretching exercises and supine rest, while the control physical activity in our study consisted of moderate intensity walking—a very common form of physical activity in pregnancy. We observed significant effects of yoga on HRV in all trimesters of pregnancy. This could suggest that regular prenatal yoga training will result in positive long-term adaptations in HRV, given the longitudinal follow-up measurements in women who practiced yoga regularly throughout pregnancy. This is further corroborated by the marked difference in ln (LF/HF) between yoga and control groups in the third trimester (Figure 1), as a physiological shift towards higher sympathetic activation towards the end of pregnancy has been previously described [26]. 

We did not find a statistically significant difference between the two study groups (yoga vs. controls) in terms of cardio-respiratory synchronization indexes. However, a trend, which did not reach statistical significance, toward higher synchronization between respiratory frequency and RR interval comparing measurements before vs. after practicing yoga was observed, with no such effect on cardio-respiratory synchronization after walking. Therefore, results on HRV changes suggesting higher vagal activity following prenatal yoga practice are at least in part supported also by cardio-respiratory synchronization analysis. This can provide an insight on mechanisms through which practicing yoga during pregnancy results in improved perinatal outcomes, such as observed reduced incidence of preterm birth, and hypertensive disorders [11,13,14,15,16]. One of the potential advantageous effects of prenatal yoga seems to be its beneficial effect on the autonomous nervous system, which contributes significantly to the physiological adaptation to haemodynamic changes during pregnancy. Our study suggested a more profound effect of yoga compared to walking on the parasympathetic activity during pregnancy. This may be explained by a more holistic approach to health management provided by yoga compared to simple physical activity such as walking. Beside physical postures (asanas), yoga include breathing (pranayama), concentration (dharana), and meditation (dhyana) techniques, which could enhance physiological rest and promote parasympathetic activation [33]. 

Our study has several limitations that have to be considered when interpreting the results. Due to the observational nature of the study, we could not control for all potential confounding factors. Although women in the study were asked to attend yoga classes weekly, it was not possible to assess exactly how often they practiced yoga and whether they also engaged in other form of physical activity besides yoga. The same is true for the control group; women in this group were recruited if they were not involved in other prenatal exercise programs, but it was not possible to control for the amount and type of their physical activity. It is possible that a higher frequency of yoga classes could have an even more pronounced effect. Especially as we were able to observe a significant increase in parasympathetic activity with only one guided yoga session weekly. In addition, all yoga classes were led by a single yoga teacher, which makes generalizing the results difficult. 

## 5. Conclusions

Prenatal yoga is associated with a significant increase in time domain HRV parameters (SDNN and RMSSD) and a significantly decreased ln(LF/HF) HRV power ratio compared to moderate intensity walking. With regular practice, enhanced parasympathetic autonomic activation following yoga persists throughout all three trimesters of pregnancy.

## Figures and Tables

**Figure 1 jcm-11-05777-f001:**
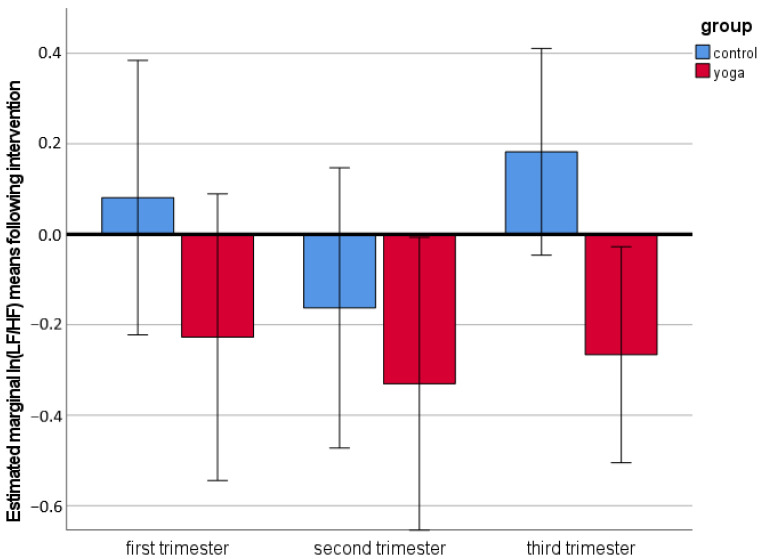
Estimated marginal means (with 95% confidence intervals) of the natural logarithm of low frequency (LF) to high frequency (HF) heart rate variability power ratio (ln(LF/HF)) following yoga and walking (control group) throughout all three trimesters of pregnancy. Note a significantly decreased ln(LF/HF) following yoga suggesting a more pronounced decrease in sympathetic to parasympathetic autonomic balance in the yoga group compared to controls (F(1,67) = 8.1, *p* = 0.006).

**Table 1 jcm-11-05777-t001:** Characteristics of study participants.

	Yoga Group (N = 33)	Control Group (N = 36)	*p*-Value
Maternal age (years)	28.7 ± 3.9	31.3 ± 3.5	0.005
Maternal height (cm)	168.2 ± 5.2	166.6 ± 5.3	0.199
Pre-pregnancy BMI (kg/m^2^)	24.06 ± 3.55	23.74 ± 3.43	0.693
Nulliparity *	26 (79%)	18 (50%)	0.001

Data are mean (standard deviation) and n (%); BMI: body mass index; * represent statistically significant difference.

**Table 2 jcm-11-05777-t002:** Heart rate variability variables before vs. after yoga practice (yoga group) or walking (control group) throughout pregnancy.

	Pre-Intervention 1st Trimester	Post-Intervention 1st Trimester	Pre-Intervention 2nd Trimester	Post-Intervention 2nd Trimester	Pre-Intervention 3rd Trimester	Post-Intervention 3rd Trimester
Heart rate (bpm) *						
Yoga	81.5 ± 9.6	74.0 ± 7.9	82.4 ± 9.4	74.9 ± 8.5	88.6 ± 10.4	80.5 ± 9.2
Controls	84.9 ± 10.3	83.5 ± 9.0	86.1 ± 9.3	83.8 ± 9.0	91.4 ± 10.3	90.1 ± 10.9
SDNN (ms) *						
Yoga	38.3 ± 13.4	46.1 ± 15.8	38.9 ± 14.4	45.7 ± 15.6	40.2 ± 16.7	44.2 ± 14.4
Controls	37.4 ± 12.8	39.0 ± 14.6	37.0 ± 12.4	37.1 ± 13.7	38.9 ± 14.4	36.3 ± 14.8
RMSSD (ms) *						
Yoga	30.5 ± 14.0	38.7 ± 14.5	30.4 ± 16.1	40.1 ± 18.5	23.9 ± 13.1	31.8 ± 17.0
Controls	29.5 ± 14.4	30.4 ± 13.9	26.8 ± 12.6	30.0 ± 15.7	23.1 ± 14.7	22.7 ± 15.6
ln(LF/HF) (−) *						
Yoga	0.05 ± 0.91	−0.18 ± 1.16	−0.02 ± 0.98	−0.35 ± 1.19	0.23 ± 1.07	−0.04 ± 1.10
Controls	0.01 ± 0.90	0.09 ± 1.06	−0.01 ± 1.03	−0.17 ± 1.02	−0.10 ± 0.96	0.08 ± 0.84

Intervention in the yoga group consisted of 90 min prenatal yoga practice; intervention in the control group consisted of 30 min walk; SDNN Standard deviation of inter-beat intervals from which artifacts have been removed; RMSSD Root mean square of successive R-R interval differences (inter-beat intervals between all successive heartbeats); ln(LF/HF) natural logarithm of the low-frequency to high-frequency power; * represent statistically significant differences following yoga vs controls in all three trimesters.

**Table 3 jcm-11-05777-t003:** Blood pressure variables, respiratory frequency, and phase synchronization variables before vs. after yoga practice (yoga group) or walking (control group) throughout pregnancy.

	Pre-Intervention 1st Trimester	Post-Intervention 1st Trimester	Pre-Intervention 2nd Trimester	Post-Intervention 2nd Trimester	Pre-Intervention 3rd Trimester	Post-Intervention 3rd Trimester
SBP (mmHg) *						
Yoga	108.0 ± 9.1	105.1 ± 8.2	106.9 ± 9.2	105.3 ± 7.0	108.6 ± 9.6	108.7 ± 8.6
Controls	109.1 ± 10.2	107.2 ± 9.1	104.9 ± 11.1	105.6 ± 10.0	110.6 ± 9.8	109.0 ± 11.3
MAP (mmHg)						
Yoga	84.7 ± 8.1	82.5 ± 6.2	83.4 ± 8.1	82.5 ± 5.6	84.9 ± 6.8	86.0 ± 6.5
Controls	85.2 ± 7.9	84.3 ± 6.6	82.0 ± 8.9	82.0 ± 7.5	86.9 ± 7.9	85.1 ± 9.4
DBP (mmHg)						
Yoga	68.3 ± 7.3	66.6 ± 5.6	67.1 ± 7.6	66.6 ± 5.5	69.2 ± 5.7	70.5 ± 5.9
Controls	69.2 ± 7.1	68.6 ± 5.8	66.5 ± 7.9	65.6 ± 6.1	71.1 ± 6.9	69.7 ± 8.7
RF (min^−1^)						
Yoga	15.6 ± 3.0	15.3 ± 3.3	15.5 ± 2.9	15.6 ± 3.1	16.4 ± 3.0	16.4 ± 3.1
Controls	17.3 ± 3.3	17.2 ± 3.6	16.4 ± 2.7	16.6 ± 2.9	16.4 ± 2.9	16.8 ± 2.8
*γ_SBP×RR_ (−)*						
Yoga	0.70 ± 0.19	0.70 ± 0.19	0.61 ± 0.22	0.62 ± 0.27	0.60 ± 0.22	0.63 ± 0.24
Controls	0.62 ± 0.23	0.60 ± 0.26	0.64 ± 0.21	0.63 ± 0.25	0.62 ± 0.20	0.60 ± 0.24
*γ_RF×RR_ (−)*						
Yoga	0.73 ± 0.17	0.74 ± 0.20	0.70 ± 0.20	0.74 ± 0.23	0.64 ± 0.27	0.69 ± 0.23
Controls	0.67 ± 0.24	0.63 ± 0.28	0.71 ± 0.20	0.69 ± 0.25	0.67 ± 0.23	0.67 ± 0.20
*γ_RF×SBP_ (−)*						
Yoga	0.78 ± 0.18	0.77 ± 0.23	0.66 ± 0.21	0.69 ± 0.29	0.70 ± 0.24	0.66 ± 0.26
Controls	0.66 ± 0.26	0.59 ± 0.30	0.69 ± 0.22	0.64 ± 0.29	0.62 ± 0.24	0.62 ± 0.28

Intervention in the yoga group consisted of 90 min prenatal yoga practice; intervention in the control group consisted of 30 min walk; SBP systolic blood pressure; MAP mean arterial pressure; DBP diastolic blood pressure; RF respiratory frequency; γ_SBP×*RR*_ synchronization index between systolic blood pressure and RR interval (inter-beat interval between all successive heartbeats); γ_RF×*RR*_ synchronization index between respiratory frequency and RR interval; γ_RF×*SBP*_ synchronization index between respiratory frequency and systolic blood pressure, * represent statistically significant differences following yoga vs. controls in all three trimesters.

## Data Availability

The data presented in this study are available on request from the corresponding author.

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
