# Peer review of "Effect of Prenatal Yoga on Heart Rate Variability and Cardio-Respiratory Synchronization: A Prospective Cohort Study"

_jcm, 2022, doi:10.3390/jcm11195777_

Round 1
Reviewer 1 Report
This is an interesting study that further supports the benefits of yoga in pregnancy. I do have a couple comments that should be addressed:
Line 41-43 (and line 287): It is incorrect to say that physical activity has been proven to reduce the incidence of intrauterine foetal growth restriction (IUGR). This has not been shown.
Line 90-91 and Table 1: It is stated that the baseline characteristics were similar in the 2 groups; however, according to the table, age was significantly different with a p value of 0.005.
Lines 242-244 and Table 3: It is difficult to see how SBP is significantly reduced post-intervention when the values are so similar (3rd trimester pre/’post are 108.6 and 108.7)?
Author Response
REVIEWER 1
COMMENT#1: Line 41-43 (and line 287): It is incorrect to say that physical activity has been proven to reduce the incidence of intrauterine foetal growth restriction (IUGR). This has not been shown.
Statements on effects of physical activity during pregnancy on IUGR have been removed.
COMMENT#2: Line 90-91 and Table 1: It is stated that the baseline characteristics were similar in the 2 groups; however, according to the table, age was significantly different with a p value of 0.005.
We are thankful to the reviewer for his remark. The paragraph describing comparison of group characteristics has been rewritten.
COMMENT#3: Lines 242-244 and Table 3: It is difficult to see how SBP is significantly reduced post-intervention when the values are so similar (3rd trimester pre/’post are 108.6 and 108.7)?
As described in the methods section, we used 2 x 3 x 2 three-way analyses of variance for determining whether a change in a given parameter (e.g. SBP) was significantly influenced by the “type of intervention” (yoga vs walking) . Time (before and after yoga or walk) and trimester of pregnancy (first, second and third trimester) were the within-subjects factors in these analyses, and intervention (yoga vs control) the between-subject factor. Even when considering physiological changes in SBP throughout pregnancy and before vs after any physical activity, SBP was more likely to be lower after yoga practice compared to walking. This has now been clarified in the methods section.
Please see the attachment.

Reviewer 2 Report
The authors admit that they could not find a statistically significant difference between the two study groups in terms of cardio-respiratory synchronization indexes. However, in the conclusion they claim to have found significant differences.
The method of continous bloos pressure measurement "derived from the finger" remains very unclear.
Overall a weak paper with weak results,
Author Response
COMMENT#1: The authors admit that they could not find a statistically significant difference between the two study groups in terms of cardio-respiratory synchronization indexes. However, in the conclusion they claim to have found significant differences.
We agree with the reviewer that statistically non-significant trends should not be over-interpreted. Nevertheless, our conclusions are based purely on statistically significant changes in HRV parameters observed: ”Prenatal yoga is associated with a significant increase in time domain HRV parameters (SDNN and RMSSD) and a significantly decreased ln(LF/HF) HRV power ratio compared to moderate-intensity walking. With regular practice, enhanced parasympathetic autonomic activation following yoga persists throughout all three trimesters of pregnancy.” We did not mention synchronization indexes in the conclusions. We did also not claim that synchronization between respiratory frequency and RR interval was higher following yoga. However, the p value of 0.057 is close to statistical significance and this trend corroborates statistically significant changes in HRV parameters described.
COMMENT#2: The method of continous bloos pressure measurement "derived from the finger" remains very unclear.
We have tried to clarify the methodology used for measuring continuous blood pressure in the methods section. We also provided a reference to a previously published article in which this methodology is described in detail and validated.
Please see the attachment.

Reviewer 3 Report
This is a welll-designed work and the paper is well-written.
There are minor spelling mistakes Line 33 hearth to be heart
The following questions should be addressed:
1. Why only one day per week yoga practice was undertaken, when usually minimum three days per week yoga practice is advised.
2. In the discussion: authors should try to explain what could be the reason that Satyapriya et al showed no changes in HRV after 16 weeks of yoga training (Line 270)
3. When there was no statistically significant difference between the two study groups in terms of cardiorespiratory synchronization indexes (Line 278-9), AFT parameters such as HR, SDNN, RMSSD and ln LF:HF were significantly different. How do the authors explain this? As this hypothesis has not been substantiated (mere increase should not be sufficient without statistical significance), lines 285-287 claiming to provide an insight into the mechanism appears an overstatement of the findings.
4. Similarly lines 290-92 stating increased parasympathetic reducing inflammation and perinatal complications are not related to the findings of the present study, does not appear relevant to write it under discussion.
5. One or two sentences should have addressed why moderate intensity walking did not bring in the similar effects as yoga for the benefit of first time readers of this article.
Author Response
COMMENT#1: Line 33 hearth to be heart.
Corrected.
COMMENT#2: Why only one day per week yoga practice was undertaken, when usually minimum three days per week yoga practice is advised.
We absolutely agree that a higher frequency of yoga classes could have had an even more beneficial effect. Especially as we were able to observe a significant increase in vagal activity with only one yoga guided session weekly. Due to logistic reasons, we were not able to conduct a study with a more intense yoga practice protocol. Future research should definitively be focused also on determining the potentially “dose-dependent” effects of yoga on ANS. Some of these considerations have now been added to the Discussion section.
COMMENT#3:In the discussion: authors should try to explain what could be the reason that Satyapriya et al showed no changes in HRV after 16 weeks of yoga training (Line 270)
There are several possible explanations for lack of long-term effects of yoga on HRV in the study by Satyapriya et al. We have summarizes few in the discussion section as per reviewer’s suggestion. The paragraph now reads:”… Satyapriya et al. showed no changes in HRV after 16 weeks of yoga training. This lack of long-term effect of yoga on HRV could have been due to several reasons. Participants in this study were randomly assigned to either yoga or standard prenatal exercise group. Therefore, their approach to yoga, motivation and compliance with the exercise protocol could have been very different from women in our study, who have chosen to practice yoga by themselves. In addition, women in the Satyapriya study mostly practiced yoga alone at home while in our study women attended weekly guided classes. This could have also impacted the motivation and attitude towards yoga practice throughout pregnancy. Moreover, prenatal exercises in the control group included both stretching exercises and supine rest, while the control physical activity in our study consisted of moderate intensity walking, a very common form of physical activity in pregnancy….«
COMMENT#4: When there was no statistically significant difference between the two study groups in terms of cardiorespiratory synchronization indexes (Line 278-9), AFT parameters such as HR, SDNN, RMSSD and ln LF:HF were significantly different. How do the authors explain this? As this hypothesis has not been substantiated (mere increase should not be sufficient without statistical significance), lines 285-287 claiming to provide an insight into the mechanism appears an overstatement of the findings.
HRV parameters and cardio-respiratory synchronization indexes are somehow related, but by no means synonymous. It is, therefore, not too surprising that we observed changes in many but not all parameters studied. However, statistically non-significant trend (p=0,057) towards higher synchronization between respiratory rate and R-R interval could theoretically corroborate HRV results suggesting an increased vagal activity following yoga compared to walking. We believe we have not over-interpreted this statistically non-significant trend, as lack of statistical significance is emphasized in both results and discussion section. Moreover, we did not state that yoga increases cardio-respiratory synchronization (please see also our response to reviewer 2 above).
COMMENT#5: Similarly lines 290-92 stating increased parasympathetic reducing inflammation and perinatal complications are not related to the findings of the present study, does not appear relevant to write it under discussion.
We agree with the reviewer. These statements have been removed.
COMMENT#6: One or two sentences should have addressed why moderate intensity walking did not bring in the similar effects as yoga for the benefit of first time readers of this article.
The following paragraph has been added to the discussion:”Our study suggests a more profound effect of yoga compared to walking on the parasympathetic activity during pregnancy. This may be explained by a more holistic approach to health management provided by yoga compared to simple physical activity such as walking. Beside physical postures (asanas), yoga includes breathing (pranayama), concentration (dharana) and meditation (dhyana) techniques, which could enhance physiological rest and promote parasympathetic activation.”
Please see the attachment.
